# The Novel Genetic Background of Infectious Bursal Disease Virus Strains Emerging from the Action of Positive Selection

**DOI:** 10.3390/v13030396

**Published:** 2021-03-02

**Authors:** Anna Pikuła, Anna Lisowska, Agnieszka Jasik, Lester J. Perez

**Affiliations:** 1Department of Poultry Diseases, National Veterinary Research Institute, 24-100 Puławy, Poland; anna.lisowska@piwet.pulawy.pl; 2Department of Pathology, National Veterinary Research Institute, 24-100 Puławy, Poland; agnieszka@piwet.pulawy.pl; 3Department of Clinical Veterinary Medicine, College of Veterinary Medicine, University of Illinois at Urbana-Champaign, Urbana, IL 61802, USA; ljperez@illinois.edu; 4Virus Discovery Group, Abbott Diagnostics, Abbott Park, IL 60064, USA

**Keywords:** very virulent infectious bursal disease virus, genetic evolution, reassortment, in vivo pathogenicity

## Abstract

The circulation in Europe of novel reassortant strains of infectious bursal disease virus (IBDV), containing a unique genetic background composition, represents a serious problem for animal health. Since the emergence of this novel IBDV mosaic was first described in Poland, this scenario has become particularly attractive to uncover the evolutionary forces driving the genetic diversity of IBDV populations. This study additionally addressed the phenotypic characterization of these emergent strains, as well as the main features affecting the viral fitness during the competition process of IBDV lineages in the field. Our results showed how different evolutionary mechanisms modulate the genetic diversity of co-existent IBDV lineages, leading to the error catastrophe effect, Muller ratchet effect, or prevalence, depending on their genetic compositions. We also determined that the action of the positive selection pressure, depending on the genomic segment on which it is acting, can drive two main phenotypes for IBDV: immune-escaping strains from the selection on segment A or strains with functional advantages from the selection on segment B. This last group seems to possess an increased fitness landscape in the viral quasispecies composition, presenting better adaptability to dissimilar environmental conditions and likely becoming the dominant population. The reassortant strains also exhibited a lower mortality rate compared with the well-known vvIBDV strains, which can facilitate their spreading.

## 1. Introduction

Infectious bursal disease virus (IBDV) is the typical member of the *Birnaviridae* family, genus *Avibirnavirus* [1], and is recognized as the causative agent of an immunosuppressive disorder known as Gumboro disease or infectious bursal disease (IBD) [2]. IBDV has the capacity to replicate into B-lymphocytes [3], leading to its depletion in the bursa of Fabricius and resulting in a prolonged immunosuppression [4]. Thus, IBDV impacts the poultry industry, causing either direct economic losses due to the mortality in the flocks or indirectly by the burden generated by the immunosuppression in the animals affected, which facilitates the infections with secondary pathogens [5], increases in the feed–conversion ratios [6], and decreases the effectiveness of the vaccines [7].

The IBDV viral particle consists of an unenveloped virion, which covers a bi-segmented (A and B) double-stranded RNA genome [8]. Segment A comprises 3.2 kb nucleotides in length organized in two partially overlapping open reading frames (ORFs) [9]. The smaller ORF encodes for the non-structural viral protein VP5, which plays a role in releasing the viral progeny from the infected cells [10,11]. Meanwhile, the larger ORF encodes a precursor polyprotein that is subsequently processed into mature proteins (VP2, VP3, and VP4) by the viral serine protease (VP4) [12]. Segment B encodes the viral RNA-dependent RNA polymerase (RdRp) (VP1), which catalyzes replication and transcription [13]. However, a recent study revealed that IBDV uses a novel cap-independent mechanism of protein synthesis initiation that relies on the viral proteins VP1 and VP3, which act as a substitute for the 5’ cap structure [14]. Of all five viral proteins, VP2 is the major structural protein that builds the viral capsid and contains the antigenic domains [15]. In fact, this protein has been historically linked to changes in pathogenicity and virulence in IBDV [16,17]. However, further studies have indicated that both genomic segments shape the phenotype, virulence, and pathogenicity in IBDV strains [18,19,20], as evidenced by the emerged natural reassortants [21,22,23,24]. Indeed, the emergence of the new viral pathogenic strains during 1980 known as very virulent IBDV (vvIBDV) resulted from a concurring temporal event in which a very virulent genetic background in segment A emerged together with the acquisition of a very virulent segment B from a strain of unknown origin [25,26]. Thereafter, other molecular mechanisms driving the genetic diversity of IBDV have been unveiled, including homologous recombination [27] and heterologous reassortment (between serotype 2 and serotype 1) [28].

Recently, the classification of IBDVs based on the sequence of the hypervariable region of the VP2 protein was introduced [29]. Seven genogroups (G1–G7) were identified, according to which a classical, antigenic variant and very virulent strains were classified as G1, G2, and G3, respectively.

Current reports suggest that the global epidemiological situation regarding IBDV is mainly characterized by the emergence of novel antigenic variants, distinct or mosaic IBDV [24,30,31]. In a more recent study, Pikuła et al. [32] uncovered the emergence of novel reassortants in Europe containing the segment A from very virulent (G3, vvIBDV-A) strains and segment B from an unidentified source, phylogenetically segregated and classified as transitional lineage (transIBDV-B). Although the potential threat that these new genetically reorganized strains represent for epidemiological control of IBD was documented by Pikuła et al. [32], the molecular, biological, and phylodynamic characteristics of these novel strains remain unclear. Therefore, the present study aimed to determine the molecular characteristics, virulence properties, ancestrality, and evolutionary forces that drive the phylodynamic behavior of the novel reassortant strains vvIBDV-A/transIBDV-B in comparison with the well-known vvIBDV lineage. We also shed light on the potential of this new emergent linage to be established and replace the previous vvIBDV lineage (vvIBDV-A/vvIBDV-B), becoming the prevalent one and being able to escape the current immunoprophylaxis policy against IBDV.

## 2. Materials and Methods

### 2.1. Sequence Dataset

All available IBDV sequences of A and B segments and the hypervariable region of VP2 (HVR), downloaded from GenBank on 30 November 2019, were analyzed. After removing poor quality and redundant sequences, a total of 172 sequences for the whole VP2 protein of segment A and 185 sequences for segment B were included in the study. The classification for the Polish IBDV strains established in Pikuła et al. [32] was kept in the current study in order to warrant homogeneity in the analyses.

### 2.2. Recombination Selection Analysis

Searches for recombinant sequences and crossover regions were performed using Geneconv, RDP, MaxChi, Chimera, BootScan, SiScan, 3Seq, LARD, and Phylpro, all implemented in RDP4 Beta 4.97 [33]. Programs were executed with modified parameter settings determined according to the guidelines in the RDP4 manual for the analysis of divergent sequences (available upon request). Recombinant sequences were tested with the highest acceptable *p* value of 0.05, and Bonferroni’s multiple comparison correction was used. Only sequences detected by more than two methods were considered recombinants. In addition, phylogenetic trees based on neighbor joining (NJ) were computed using the segments identified as parental. All sequences detected as recombinant were removed from the evolutionary analysis.

### 2.3. Evolutionary History and Population Dynamics

To estimate the time of the most recent common ancestor (tMRCA), we followed the methodology described by Alfonso-Morales et al. [34]. Briefly, all the sequences selected from the dataset were used to generate the BEAST input file by BEAUti within the BEAST package v1.10.4 [35]. The tMRCAs were estimated by employing a Bayesian MCMC approach. The Bayesian estimating model marginal log likelihood through the path sampling (PS) and stepping stone (SS) sampling methods was used. In addition, a Bayesian skyline plot (BSP) for IBDV strains to infer the population dynamics in terms of changing levels of relative genetic diversity (Neτ) through time was performed. For BSP analysis, data were collected and plotted using GraphPad Prism software 8.4.1 (1992–2020, GraphPad Prism software LLC). In all cases, the Markov chain Monte Carlo (MCMC) chains were run for 2.5 × 10^8^ generations to obtain an ESS > 250, and the first 10% of the trees were discarded as burn-in. Finally, trees were visualized and edited in FigTree. For segment A, the strain JQ411012 was used as an outgroup, and for segment B, the group of sequences from the lineage non-vvIBDV was used to root the three to infer the ancestrality for the other two identified lineages.

### 2.4. Positive Selection Analysis

The hypothesis of positive selection on the VP2 and VP1 genes of IBDV was tested by the site models and branch site models implemented in the CODEML program of the PAML v4.9 software package [36], following the criteria reported by Rios et al. [37]. Briefly, different values of the non-synonymous/synonymous dN/dS rate ratio (ω parameter) were assessed, and false positives were avoided by contrasting the models used to detect sites under positive pressure with models used to detect neutral selection [36,38]. The Bayes Empirical Bayes (BEB) calculation of posterior probabilities for site classes was used to estimate the probabilities of sites under positive selection. The VP1 gene was also tested for a discrete distribution using M3 and M0 models. Model M0 allows for a single ω value across the whole phylogenetic tree at all sites, and M3 assumes multiple categories of selection, not necessarily positive selection. Subsequent models allow ω to vary at different sites.

Branch site tests, using pre-specified branches, are hypothesized to have occurred (foreground branches), and were made with the null Model A1. This allows ω ratios to vary among sites and among lineages, and it also provides two additional classes of codons with ω = 1 along pre-specified foreground branches, while restricting ω to be ≤1 on background branches. The alternative Model A allows ω to vary between 0 and 1 and be equal to 1 for all branches, and has two additional classes of codons under positive selection, with ω > 1 along pre-specified foreground branches, while restricting ω to either 0–1 or ω = 1 on background branches.

For all likelihood ratio tests (LRTs), the null model is a simplified version of the selection model with fewer parameters, and is thus expected to provide a poorer fit to the data (lower maximum likelihood). The significance of the LRTs was calculated assuming that twice the difference in the log of maximum likelihood between the two models was distributed as a χ^2^ distribution with the degrees of freedom (df) given by the difference in the number of parameters in the two types of models [37].

The substitution rate ratios of non-synonymous (dN) versus synonymous (dS) mutations (ω) were calculated. In the site model, codon site models M1, M2, M7, and M8 were implemented using the LRT to evaluate whether the site models assuming positive selection (M2 and M8) fit the data better than models without positive selection (M1 and M7). To test whether codon selection occurs among different IBDV lineages, the branch site Model A, which allows for ω > 1 along foreground branches, was compared with the null Model A1, which only allows for ω ≤ 1 along foreground and background branches. This model comparison was made by labelling foreground branches of the three main clades A1–A3 for the VP2, vvIBDV, and transIBDV for the Polish strains.

### 2.5. Functional Analysis

Type I of functional divergence detects differential rates (acceleration or deceleration) in each amino acid position, whereas Type II describes a burst of rapid evolution immediately after gene duplication [39]. The software DIVERGE 3.0 was used to detect the functional divergence among different clades and lineages of Polish IBDV strains, as described in Rios et al. [37]. Briefly, coefficients of Type I and Type II of functional divergence (θI, θII) were calculated. If the coefficients are significantly supported, it means that site-specific altered selective constraints or a radical shift of the amino acid physiochemical property occurred after the diversification of the lineage. Moreover, a site-specific posterior analysis was used to predict amino acid residues that were crucial for functional divergence. In this analysis, large posterior probability (Qk) indicates a high possibility that the functional constraint (or the evolutionary rate) and/or the radical change in the amino acid property of a site is different between two clusters.

### 2.6. D Protein Visulization

To visualize all of the mutation and the sites under different evolutionary forces, the 3D crystal structures of VP2 and VP1 proteins were obtained from the Protein Data Bank (PDB). Crystals obtained with lower values of resolution for each protein were downloaded, thus, the selected accession numbers were used, VP2 protein (PBD:2DF7) and VP1 protein (PDB:2PGG). In all cases, to simplify the visualization, monomers and spheres were used to denote the sites and colors for domain. In all cases, the software package PyMOL Molecular Graphics System Version 2.0 (Schrödinger, LLC) was used.

### 2.7. Experimental Study

#### 2.7.1. Ethical Statement

All of the methods and procedures linked with the current study were supervised and approved by the Local Ethical Commission in Lublin (Permit No. 82/2015), in agreement with the rules in place in the EU (Directive 2010/63/UE).

#### 2.7.2. Viral Samples

Three IBDV strains, Bug/03/Poland/2003, 117/14/Poland/2014, and 75/11/Poland/2011, selected from a total of 46 Polish IBDV field strains, collected during 1991 to 2015, and previously classified by Pikuła et al. [32], were used to estimate virulence and pathogenesis in an animal trial. The selection of the strains was based on the fact that the strains Bug/03/Poland/2003 and 117/14/Poland/2014 were genetically classified as novel reassortant strains containing a unique genetic background [32], and the strain 75/11/Poland/2011 was biologically characterized previously and classified as a very virulent reference strain by Pikuła et al. [24]. The viral titre of each strain was determined in 10-day-old, specific pathogen free (SPF) embryonated eggs via the chorioallantoic membrane route (CAM) and expressed as an embryo infective dose/mL (EID_50_/mL).

#### 2.7.3. Sequencing of the Full-Length A and B Segments of IBDV Strains Bug/03, 117/14, and 75/11

In order to obtain the complete genome (segments A and B) of the IBDV strains used in the experimental study, the viral RNA was extracted from bursal samples using IndiSpin Pathogen Kit (Indical Bioscience) according to manufacturer instructions. The full length for both segments of the IBDV viral genome was obtained by using the SuperScript III One-Step RT-PCR System with High Fidelity DNA Polymerase (Invitrogen) according to the manufacturer. The primers used in this study were designed based on the nucleotide sequence of the D6948 strain (GenBank Accession numbers AF240686 and AF240687 for segment A and B, respectively) and are available upon request. The obtained amplicons after purification with the QIAquick Gel Extraction Kit (Qiagen) were sequenced in both directions by a commercial service (Genomed, Poland). Nucleotide sequences were manually assembled using MEGA v6.0 [40], and the consensus sequences were submitted to GenBank (Accession No. MT629830 – MT629835).

#### 2.7.4. Pathogenicity Study in Chickens

A total of 40 five-week-old SPF chickens (White-Leghorn, Valo BioMedia, Germany) were randomly divided into four groups containing 10 birds/group (Groups A–D). Birds of groups A–C were inoculated intraocularly with 10^5^ EID_50_/mL of Bug/03/Poland/2003, 117/14/Poland/2014, and 75/11/Poland/2011, respectively, whereas chickens from group D were inoculated with a saline buffer solution (PBS 1X) and used as the control group. All groups were separately housed in HEPA-filtered isolators (Montair Andersen B.V., Holland) and monitored daily for 10 days. All dead or euthanized birds were examined at necropsy, and the bursae of Fabricius (BF) were taken for further analysis, including histopathology and the determination of the bursa to bodyweight index (B-BW) [28].

#### 2.7.5. Histopathology

The histological preparations of collected BF were performed as previously described [24]. Briefly, each bursa sample was individually examined and evaluated for histopathological abnormalities, which were scored according to the criteria described by Jackwood et al. [28].

#### 2.7.6. Statistical Analysis

The obtained B-BW of each experimental group and the histological scoring were compared using a Brown–Forsythe and Welch ANOVA for multiple comparison with a Dunnett’s T3 multiple comparisons test correction implemented in Graphpad Prism software 8.4.2 (1992–2020, Graphpad Prism Software Inc.). The results with *p* < 0.05 were considered as statistically significant.

## 3. Results

### 3.1. Molecular Characterization of VP2 and VP1 Proteins in All of the Polish IBDV Strains Studied

The analysis of the deduced amino acid sequences of all Polish IBDV strains revealed the presence of four residues within the HVR, which are characteristic of very virulent strains (G3a, ^222^A, ^256^I, ^294^I, ^299^S). In addition, the presence of ^324^L was found to be characteristic of the A3 lineage (Figure 1). Interestingly, this amino acid is located in the major hydrophilic peak P_HI_ (312–324) of HVR, and has been linked to the loss of reactivity to the monoclonal antibody (Mab 8) [41,42]. Similarly, all Polish IBDV strains from the A1 lineage presented a ^270^T, which is considered a signature of antigenic variants or classical IBDVs [43]. The rest of the exhibited amino acid changes are presented in Figure 1.

Regarding segment B, it was revealed that the Polish strains included in the vvIBDV-B lineage showed an amino acid profile typical for this lineage (Figure 1). In the case of the transIBDV-B, besides the pattern reported in Pikuła et al. [32], which is kept for all of the strains of this novel lineage, two additional alterations in the viral polymerase were present, ^233^LxI and ^263^VxI (Figure 1).

### 3.2. Recombination on the VP1 Gene of IBDV Is a Mechanism of Genetic Variability on Polish Strains

From the homologous recombination analysis on the whole VP2 gene, three sequences were identified as recombinant, however, none of them belonged to the Polish IBDV strains. Meanwhile, the analysis of VP1 yielded eight strains detected as homologous recombinant strains. Within this group, the strain 75/01K4/Poland/2001 (Genbank ID: KX759539) was identified to be the result of the recombination event, with the strains Variant E/USA/1985 (Genbank ID: AF133905) and the Polish strain 75/01K7/Poland/2001 (Genbank ID: KX759540) as parental strains (Appendix A).

### 3.3. Temporal Reconstruction, Evolutionary Rates, and Population Dynamics of Polish IBDV Strains

A recent study reported the phylogenetic segregation of the IBDV strains circulating in Poland for both segments forming different clusters [32]. Thus, for segment A, three main clusters within the genogroup 3a containing only IBDV Polish strains were identified [32], whereas, for segment B, two main lineages were observed: a very virulent lineage and a novel lineage classified as transitional B lineage (transIBDV-B) [32]. The temporal analysis for all IBDV Polish lineages for both segments revealed different ancestrality for each main monophyletic group (Figure 2). For segment A, the clade previously classified as A3 [32] was identified as the most ancestral, with a tMRCA in 1983 (95% HDP: from 1980 to 1991) (Figure 2A). In this clade were grouped strains that were circulating in Poland during the years 1991 to 2010. The clade A1 showed a tMRCA in 1988 (95% HDP: from 1984 to 1992), with strains that caused IBD outbreaks from 1992 to 2015, indicating that strains from this clade have continuously circulated in the field (Figure 2A). Meanwhile, the clade A2 was identified as the most recent one, with a tMRCA located in 1997 (95% HDP: from 1989 to 2000) and strains that circulated during 2000 to 2015 (Figure 2A).

Like segment A, the temporal analysis for segment B also revealed different dates for the ancestors of the circulating IBDV Polish strains (Figure 2B). For the very virulent B lineage (vvIBDV-B), three main clades were identified with tMRCA between 1985 and 1987, suggesting a potential common origin or emergence for the strains that belonged to this lineage (Figure 2B). In the case of the recently classified transIBDV-B lineage, the tMRCA was determined in around 1981 (95% HDP from 1978 to 1992), with a consequent diversification in two main clades that emerged in 1990 (95% HDP: from 1984 to 1995) and 2006 (95% HDP: from 1997 to 2011) (Figure 2B).

Interestingly, the evolutionary rates for each lineage for both segments yielded different values (Figure 2A,B). For segment A, the evolutionary rate of the strains included in the clade A3 showed a lower mean value than other lineages (Mr = 8.2 × 10^−5^ substitutions/site/year), which could be linked to the probable extinction of strains from this clade (since, after 2010, there is a lack of detection of the strains from this group). The strains located in the clades A1 and A2 showed similar evolutionary rates (Mr(A1) = 4.0 × 10^−4^ substitutions/site/year and Mr(A2) = 2.09 × 10^−4^ substitutions/site/year), with a slightly higher value for the strains located in the A1 clade, which is reflected by the higher diversification into this group observed by the higher number of bifurcation in the branches (Figure 2A). For segment B, the evolutionary rates for both main groups of Polish IBDV strains showed similar values (Figure 2B). Thus, for the vvIBDV lineage, the Polish strains showed an Mr = 4.2 × 10^−4^ substitutions/site/year, whereas the new transIBDV lineage showed an Mr = 2.4 × 10^−4^ substitutions/site/year.

### 3.4. Positive Selection and Functional Divergence Act on Segments A and B of the Polish IBDV Strains, Inducing Evolutionary Advantages

From the clades A1–A3 for the VP2 lineages tested, the results obtained showed that the action of positive selection caused the emergence of the three IBDV Polish clades, with values of ω^2^ = 217.4730 for clade A1, ω^2^ = 1.6607 for clade A2, and ω^2^ = 216.4132 for clade A3 (*p* < 0.01) (Figure 3 and Appendix A). On the lineages A2 and A3, no sites were identified that could represent an evolutionary advantage, whereas, on the lineage A1, 17 sites were found (*p* < 0.01) (Appendix A) within the hypervariable domain of VP2, of which four sites were on the P_BC_ loop, one site was on each P_DE_ loop and P_HI_ loop, and two sites were on the P_FG_ loop (Figure 3). In the case of VP1 lineages, non-vvIBDV, vvIBDV, and transIBDV, where the Polish IBDV strains were located, the branch site selection was tested. The results revealed that the action of positive selection caused the emergence of the transIBDV lineage, with a value of ω^2^ = 16.3423 and a statistical support of *p* < 0.01, and there was no additional evidence of the action of this evolutionary force on the other two lineages (Figure 3 and Appendix A). On the transIBDV lineage, two sites were found (*p* < 0.01) under the action of positive selection, the codon 396 located on Motif C of the *palm* and the codon 466 located on the fingers domain 2 (Figure 3 and Appendix A).

The hypothesis of positive selection on the VP2 and VP1 genes of IBDV was also tested by the site models implemented in the CODEML program of the PAML v4.9 software package [36]. Based on the Bayesian posterior probabilities, four codon sites for VP2 (positions: 217, 248, 249, and 315) under positive selection pressure were identified from the M2 and M8 models (Appendix A). All four codon sites were selected with a 0.01 significance level (Appendix A). Analyzing the location of the sites, they were situated on HVR, one being in the P_BC_ loop, two in the P_DE_ loop, and one in the P_HI_ loop (Figure 4A). None of the codons were identified under positive selection or discrete selection by the site model for VP1.

From the functional divergence analysis, it was obtained that none of the codons for VP2 were selected for Type I or Type II functional divergence in any of the three A1–A3 clades for the Polish IBDV. In the case of VP1, a total of five sites were predicted through Type II functional divergence analysis (Appendix A). To examine the potential role of these amino acid changes among the different segment B from the Polish IBDV lineages, the sites identified were mapped on the VP1 protein structure (Figure 4B). Two sites were identified in the *fingers* domain (189 and 212), which is linked to capturing and binding the rNTPs to the nascent strand of the viral genome. Moreover, three sites were located on the N-terminal domain, located on the groove involved in the stabilization of the non-template RNA strand during the initiation of the transcription (Figure 4B).

### 3.5. Evaluation of Virulence and Molecular Signatures Linked to the Biological Properties of the Novel Re-Assortant Strains vvIBDV-A/transIBDV-B

To unravel all of the information that could be potentially linked with the virulence and biological properties of the novel reassortant strains vvIBDV-A/transIBDV-B, the deduced amino acid sequences obtained from the complete genome of the strains used in the experimental infection were inspected. All amino acid replacements that were located on the mature proteins were considered non-relevant (Appendix A), whereas all of the molecular signatures presented on the immature viral proteins were mapped on the 3D structures (Figure 5A). Thus, for the ORF1 (VP5) of segment A, when compared to 75/11/Poland/2011, both reassortant strains Bug/03/Poland/2003 and 117/14/Poland/2014 presented two amino acid differences (^14^ExK and ^65^IxV), which are located in a loop of the N-terminal domain (Figure 5A) and common to all IBDV strains. Notoriously, the strain 117/14/Poland/2014 showed two additional replacements, ^101^PxT and ^105^GxS, the first one being quite relevant, since it represents a mutation in one of the three poly-proline motifs (P**P) (Figure 5A) related to the activation of the PI3K/Akt pathway [44]. In the VP2 encoded by ORF2, the reassortant strains Bug/03/Poland/2003 and 117/14/Poland/2014 have a ^270^AxT replacement present in the attenuated strain located on the B-sheet of the HVR (Figure 5A). Additionally, the strain 117/14/Poland/2014 presented a replacement ^79^NxS in the Sc’Sc’’ loop (Figure 5A), however, these positions have not been previously linked to any functional role of VP2. In segment B, a total of 12 amino acid replacements were observed for the reassortant strain 117/14/Poland/2014 and 11 for the strain Bug/03/Poland/2003 (Figure 5A). In both strains, three replacements, ^145^TxS, ^146^DxE, and ^147^NxG, were located in the N-terminal domain of VP1 (Figure 5A). Whereas the replacement ^145^TxS has been characteristic of reassortant strains, the changes ^146^DxE and ^147^NxG are distinctive in attenuated strains. Moreover, both strains presented the replacements ^219^DxE and ^242^ExD in the *finger* domain I (Figure 5A). The 117/14/Poland/2014 strain also presented in the *finger* I domain of VP1 the replacement ^233^LxI. In addition, another four signatures from classical and attenuated IBDV strains were presented in the VP1 of both reassortant strains, including ^393^DxE and ^562^PxS located in the *palm* domains I and II, respectively, and ^687^PxS and ^695^RxK both located in the C-terminal domain (Figure 5A). A notorious and unique molecular signature presented in both strains was observed with the replacement ^689^SxN.

From the experimental infection in SPF chickens, for both reassortant strains Bug/03/Poland/2003 and 117/14/Poland/2014 (groups A and B), a 20% and 30% mortality, respectively, was observed (Figure 5B). In comparison, the animals infected with the reference very virulent IBDV strain 75/11/Poland/2011 (group C) showed a 50% mortality. No mortality was observed in the animals from the control group (group D). From the evolution of estimated B-BW ratios, a similar degree of bursal atrophy in the animals infected with the reassortant strains Bug/03/Poland/2003 and 117/14/Poland/2014 was observed, with the vvIBDV strain 75/11/Poland/2011 showing no statistical differences among these groups (groups A–C) (Figure 5C). Moreover, these three groups showed statistically lower B-BW ratios when compared with the control group (group D) at 10 dpi (Figure 5C). From the clinical evaluation, the animals from groups A–C presented typical clinical signs compatible with IBD, including ruffled feathers, depression, and watery diarrhea. At the end of the experimental infection (10 dpi), the BF exhibited severe atrophy (Figure 6C,D), the overall histological scoring obtained from the examination of the collected BF indicated that both reassortant strains and the very virulent strain caused severe microscopic lesions without statistically significant differences among them, and no histological lesions were found in the control group (Figure 5D). At the time of necropsy (10 dpi), no clinical symptoms or gross bursal lesions were found in the animals from the control group (Figure 6E). In the animals from groups A–C, ecchymotic hemorrhages were observed in the enlarged BF and the mucosal lining of the proventriculus, and the kidneys were swollen (Figure 6). In the early phase of infection (3–5 dpi), the BF was infiltrated by inflammatory cells (heterophils and macrophages) and displayed lymphocyte depletion or necrosis of medullary and cortical regions of the bursal follicles (Figure 7C,E). At 10 dpi, necrosis, atrophy, or fibrosis of lymphoid follicles were observed in the bursal tissue (Figure 7D,F).

## 4. Discussion

IBDV is linked to one of the most economically devastating diseases in the poultry industry. Despite the prophylactic vaccination and biosecurity measures applied globally, the virus remains endemic worldwide [45]. This last issue highlights the current need for a continued molecular surveillance policy to track the emergence of new variant strains of IBDV and uncover the molecular mechanism driving the genetic diversity of this viral pathogen. A recent report by Pikuła et al. [32] denoted the segregation of the Polish IBDV strains in three different lineages from the genogroup 3a (G3a) for segment A and two main lineages for segment B (very virulent and a unique transitional lineage), suggesting a high degree of genetic diversity in the strains of IBDV circulating in Polish flocks. Since this country has applied and updated several control programs against IBDV, including the vaccination of progeny with an intermediate vaccine (since 1993) [46] and introduction of new intermediate plus (first in 1996 and six more between 1999–2010) and vector (2008) vaccines, the emergence and establishment of novel lineages able to evade the control measures and compete with previously established IBDV strains makes this scenario particularly attractive to gain a deep understanding of the evolution, diversity, and pathogenicity of IBDV. Hence, the current study addressed rational *in silico* designs and *in vivo* and *ex vivo* experiments to unravel the phylodynamic, evolutionary, and biological properties of IBDV in Poland, with special attention paid to the novel emergent reassortant lineage vvIBDV-A/transIBDV-B.

Homologous recombination events have been described as rare or with a very low frequency in IBDV populations [47]. Indeed, only two previous studies have reported recombination events in segment A [27,48], and two in segment B [47,48]. For segment A, homologous recombination has been described between very virulent strains and attenuated strains, the latter used as vaccine strains. However, since the breakpoint positions found so far did not produce antigenic variations, it is unlikely that these events have favored an escaping variant to the vaccinations used in the field [49]. For segment B, homologous recombination has been found between very virulent strains [47,48]. Concurring with these previous results, in the current study, a very virulent Polish IBDV strain 75/01K7/Poland/2001 was identified as the result of homologous recombination. Nonetheless, the functional role played by this event in this segment is still unknown.

Unlike homologous recombination, nucleotide substitutions are highly frequent events in IBDV [34]. Since IBDV is a quasispecies virus [50], the viral composition in the infected animals consists of mutant spectra [51]. Analyzing the results obtained from the molecular characterization study for the IBDV Polish population, different molecular signatures for each specific lineage were observed. In segment A, whereas the fixed mutation found in the lineage A3 is in a site that promotes escape to the immune response [41,42], the one found in lineage A1 is thought to be characteristic of non-vvIBDV strains [43]. Thus, both viral populations fixed different strategies of competition in the field, however, the strategy of the A1 lineage, known in quasispecies viruses as “suppression to keep low frequency and preservation in the population” [52], could guarantee a higher circulation of this group of strains. However, the strains from A1 with segment B from the transIBDV lineage acquired more genetic changes, indicating that strains from this particular group are actively evolving in the field.

The results obtained from the temporal analysis indicated that segment A of the Polish strains that circulated in IBD outbreaks during the 1990s (~1991) in Poland [53] (clades A1 and A3) was originated close to the event of the emergence and expansion of the very virulent (vvIBDV-A) genetic background [34]. After the vvIBDV-A emerged in Asia in 1980, it was probably introduced in Europe by migratory wild birds [34], and quickly spread in this continent by the live poultry trade. Therefore, it is likely that both lineages (A1 and A3) were introduced in Poland by different poultry importations. On the other hand, segment A from the IBDV strains that were detected circulating in the field during 2000 to 2005 (clade A2) emerged during 1997, pointing to a potential introduction from France, suggested previously by Domańska et al. [53]. Hence, the genetic diversity found in the Polish IBDV population regarding segment A is influenced by external introductions that renovated the genetic background of the IBDV strains. Nonetheless, to identify the most probable geographical origin of segment A of the Polish strains, a phylogeographic approach using geographic and phylogenetic association traits will be required. For segment B, both main lineages yielded a similar date for the ancestors of the strains circulating in Poland (1980s), suggesting that a high diversity for this segment, which encodes for the polymerase, has been kept in the Polish strains. The evolutionary rates obtained for the Polish strains for segment B are considerably high for both lineages, suggesting that these strains are in a fast-rate evolutionary process. This is a relevant aspect to denote, since quasispecies viruses rely on the diversity of their polymerases to generate broader or narrower mutant spectra [54]. It is also important to highlight that, since the transIBDV has only been detected in Poland, with the exception of Finland in 2014, it is likely that this genetic background emerged in Poland from an undetermined natural source. This hypothesis is supported by recently published results from an IBDV survey conducted in the Netherlands, Belgium, Germany, Denmark, Latvia, the Czech Republic, and Sweden, where the presence of similar reassortants has not been confirmed [55]. In addition, the emergence of strains with a transIBDV-B composition undoubtedly introduces a source of competition in the field, favoring the acquisition of a higher genetic fitness for mutant spectra of the Polish IBDV strains. This particular characteristic, together with the fact that the Polish flocks are under a prophylactic vaccination against IBDV [41,53], seems to lead the phylodynamic behavior of the IBDV in this country. From the analysis of the genetic diversity versus time, it was inferred that the phylodynamic behavior of the strains belonging to the clade A3 experienced an extinction process since 2010, and those from the clade A2 have kept their genetic diversity constant, suggesting that the vaccination programs established in Poland have successfully controlled these strains in the two groups. Indeed, this aspect can also be corroborated by the fact that the phylodynamic behavior of the lineage vvIBDV-B drastically dropped the genetic diversity over time. Contrarily, the fact that the transIBDV-B lineage has increased its genetic diversity (mainly after 2010) could be indicative of the reassortant strains containing this newly described genetic combination (vvIBDV-A/transIBDV-B) [32], and could enhance their genetic fitness. Since the strains containing the transIBDV-B genetic background were located only in the A1 clade, which has also shown a higher substitution rate for segment A, this can indicate that this novel genetic composition in segment B favors the heterogeneity of the viral population, representing an evolutionary advantage to this particular type of strain and allowing it to overcome additional barriers, such as host immunity [56,57].

Analyzing in detail the characteristics of heterogeneity within the viral population for the Polish IBDV strains, we investigated the action of positive selection and the functional divergence properties on each Polish main lineage. When site and branch site models were applied to the VP2 coding region, it was observed that the three main lineages were under positive selection, however, no sites were fixed for the A2 and A3 lineages. This failure could be a consequence of repetitive bottleneck actions generated either by the immune response of the host elicited by the vaccination or by the intra-mutant competition generated by the circulation of strains with a diverse genetic composition. Hence, these continued barriers have been imposed to those lineages, leading to the error catastrophe effect (observed in lineage A3) or the Muller ratchet effect (caused by the incapacity to generate highly diverse genetic populations, i.e., constant genetic diversity in time observed in lineage A2) [58]. Meanwhile, for the strains belonging to the A1 clade, several sites located mainly on HVR, potentially the viral escaping the immune response of the host and the viral attachment to host cells, were identified.

The main difference found among the strains from the A1 clade to the A2 and A3 clades is that most of the strains from the first one contains the novel transIBDV-B segment, which seems to play a role in facilitating the fixation of mutations induced by the positive selection pressure. This hypothesis came from the fact that, when branch site models were applied to segment B lineages, it was observed that the transIBDV-B lineage was the only one positively selected. Considering that VP1 is not a structural protein of viral capsid, the positive selection action on this lineage cannot be related with an advantage linked to the escape from the immune response from the host, but with a functional role. Looking at the results yielded by both the positive selection and the functional divergence analyses, the positively selected sites were located on the *fingers* and *palm* domains of VP1, whereas the functional divergence sites were located at the N-terminal domain related with the transcription of the viral RNA and the C-terminal domain involved in the addition of dNTPs in the nascent RNA stream [59]. Notoriously, all of these mentioned domains have been linked to the fidelity of the polymerase activity of IBDV [18]. Thus, the emergent Polish reassortant strains vvIBDV-A1/transIBDV-B probably have lower polymerase fidelity compared to the vvIBDV-A/vvIBDV-B strains. This lower fidelity could give an adaptive advantage to these types of strains. It has been recently uncovered that IBDV’s replication takes place thorough a mechanism of viral factories [60], which favors the stamping machine replication mode that generates a lower level of heterogeneity compared to the geometric replication mechanism [61]. Hence, the IBDV quasispecies cloud composition relies more on the capacity of the polymerase to generate the mutant spectra than the mechanism of replication per se. This characteristic is reflected in the mutation rate of IBDV, which has been reported, as a consensus, in the order of approximately 10^−4^ substitutions/site/year [47,62,63], which is one order of magnitude lower than most of the quasispecies viruses [64]. In addition, IBDV shows one of the higher mutation rates in the polymerase gene, as determined by Gao et al. [62], concurring with the results of the current study. This particular property is indicative of the relevance of a highly variable polymerase in IBDV to guarantee a variable progeny, which increases the fitness landscape in the viral population, despite generating defective mutants. Hence, the emergence of a particular reassortant strain carrying genetic information for the polymerase gene with lower fidelity, as seems to be the case of the novel vvIBDV-A/transIBDV-B, will generate a mutant spectrum that contains a highly heterogeneous population and is composed of some variants located at the top of a sharp fitness peak, with others considered defective variants. Thus, it is highly probable that the novel vvIBDV-A/transIBDV-B reassortant strains have a higher level of adaptability to face the current environmental conditions in the field, including the bottleneck effects generated by the vaccination policies in place and the competition process against other intra-mutant IBDV quasispecies, therefore becoming a dominant population (summary in Figure 8).

A relevant aspect related to the level of heterogeneity linked to the mutant spectra composition in RNA viruses is the virulence of the strains [65]. Several studies conducted for other RNA species under positive selection pressure have shown a temporal decrease in virulence [51,56,57]. In the current study, two of the novel reassortant strains, vvIBDV-A/transIBDV-B, were included in an experimental infection to uncover the virulence and pathogenic properties of this emergent and unique genetic combination of IBDV. In concordance with the results obtained for other positively selected RNA viruses, the novel vvIBDV-A/transIBDV-B reassortant strains showed a lower mortality percentage when compared to the very virulent Polish reference strain 75/11/Poland/2011. On the one hand, it has been reported for IBDV that positive selection of VP2 protein leads to a decrease in virulence [66,67]. On the other hand, we previously mentioned that the Polish IBDV strains have been under a constant selective pressure induced by immunoprophylactic vaccination, which produces repetitive bottleneck effects in the viral population. Thus, Coronado et al. indicated that bottleneck effects introduce in the viral population a strong element of stochasticity during the selection process, mainly in the short-term (intra-host), producing a decrease in virulence [56]. An apparent contradiction could be the results recently described by Tammiranta et al., who reported the presence of the novel Polish reassortant strains vvIBDV-A/transIBDV-B in Finland in outbreaks of IBD that started in 2014 [68]. From the evaluation of virulence, as well as the reported mortality in the field, these authors concluded that the behavior of these strains resembled very virulent strains. However, the resilience mechanism of the quasispecies allows a low virulent mutant spectrum to regain its viral fitness and fully recover the high level of virulence once the conditions that kept the quasispecies in a constrain change or are eliminated [52,54,56]. Then, it is highly probable that, with the vaccination applied in Finland, with the chicken breeds used in poultry farming and other additional factors differing from those in Poland, the novel reassortant vvIBDV-A/transIBDV-B could present a different virulence degree. Nevertheless, considering that the virulence of IBDV is a complex process that relies on the interaction among the proteins yielded by both viral genomic segments and the host cells, as well as the fact that the precise molecular mechanism responsible for its regulation remains unclear, further investigations are needed to clarify, mainly using reverse genetic approaches, the specific role of each replacement found under positive selection pressure on both segments, as well as the synergism that each replacement could play between segments linked to the virulence of these novel IBDV strains.

Notoriously, from the clinical and histological evaluation, it was observed that the intensity of the bursal atrophy found in the animals infected with the novel reassortant strains showed the same degree of abnormalities as the very virulent reference strain. This aspect could be influenced by the fact that the residues on VP2 found to be positively selected can guarantee an efficient attachment to the host cells. Therefore, from all molecular in vivo and ex vivo experiments, it can be inferred that the virulence properties of the novel reassortant strains vvIBDV-A/transIBDV-B, with the exception of mortality, are similar to the very virulent strains. This high degree of affectation found in the bursa of Fabricius is also indicative of a high level of immunosuppression that these novel strains can cause in the affected chickens. Therefore, even though these strains could cause lesser economic losses linked to direct mortality, they can still originate indirect immunosuppression-related losses just as serious or even more serious than the very virulent strains. This last premise is sustained by the fact that, since these strains induce a lower mortality, they have a higher opportunity to propagate within the host population, because the number of individuals will not drop drastically. Hence, the novel Polish reassortant vvIBDV-A/transIBDV-B strains represent a serious threat to the poultry industry.

Although the current study has uncovered relevant aspects, it is not exempt of limitations. For instance, all of the results obtained strongly suggest that the mutant spectra of the novel reassortant strains vvIBDV-A/transIBDV-B have a higher level of genetic diversity than the vvIBDV-A/vvIBDV-B strains, but to confirm this characteristic, an experimental study using next generation sequencing unveiling the quasispecies composition of these two viral populations will be required. Likewise, a second aspect in this study that remains unknown is the origin of the novel transIBDV-B genetic background of the Polish IBDV strains. There is also an urgent need to incorporate more complete sequences of both genome segments of IBDV to perform reliable phylogeographic studies that will allow the inference of the origin and transmission of emerging lineages of this viral agent. In addition, since the role of wild migratory birds seems to be significant in the epidemiology and emergence of IBDV [26,34], a bigger effort combining the results from different research groups is imperative to conduct molecular epidemiological studies targeting IBDV in wild birds.

## 5. Conclusions

The current study provides novel and significant insights into the emergence, diversification, and phylodynamic process of IBDV. The results obtained revealed the co-existence, for almost 30 years, of divergent IBDV lineages in the field, which presented different mechanisms to modulate their genetic diversity and their viral transmission. We can also conclude that the action of the positive selection pressure, depending on the genomic segment on which it is acting, can drive at least two main phenotypes in IBDV. Thus, the viral strains selected due to the pressure on segment A seem to have advantages related to the immune response of the host, and are favored during bottleneck effects induced by vaccination, whereas individuals selected on segment B can have functional advantages that guarantee a higher diversity in the mutant spectra of the viral population, increasing their fitness landscape in the viral quasispecies, presenting better adaptability to dissimilar environmental conditions, and likely becoming the dominant population. This study also highlights the emergence of the very virulent strains during 1980 in concordance with previous studies, but in particular denoted the emergence and establishment of a new genetic background for segment B of IBDV at the same time as the very virulent lineage. The fact that this new lineage emerged and competed with the well-established very virulent lineage, becoming the prevalent lineage in Poland and disseminating to other countries in Europe, represents a serious threat for the poultry industry worldwide. Our findings bring important insights into the mechanism of IBDV evolution and indicate the demand for coordinated international epidemiological investigations.

## Figures and Tables

**Figure 1 viruses-13-00396-f001:**
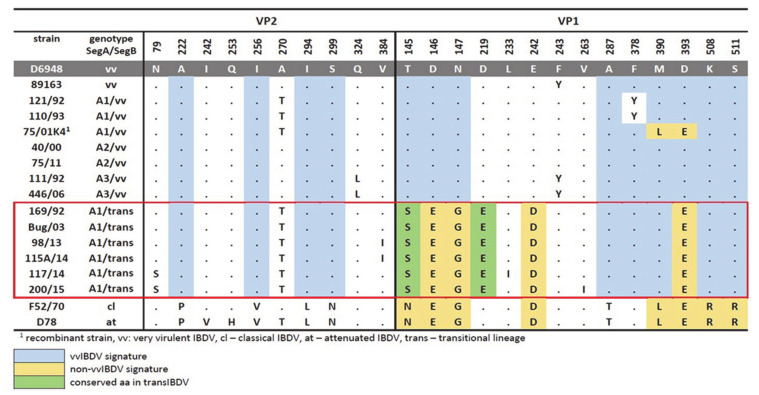
Comparison of amino acid substitutions at different positions on VP2 and VP1 proteins between Polish IBDV and reference strains. Dots show an identity with D6948. A—alanine, R—arginine, N—asparagine, D—aspartic acid, Q—glutamine, E—glutamic acid, G—glycine, H—histidine, I—isoleucine, L—leucine, K—lysine, M—methionine, F—phenylalanine, P—proline, T—threonine, Y—tyrosine, V—valine.

**Figure 2 viruses-13-00396-f002:**
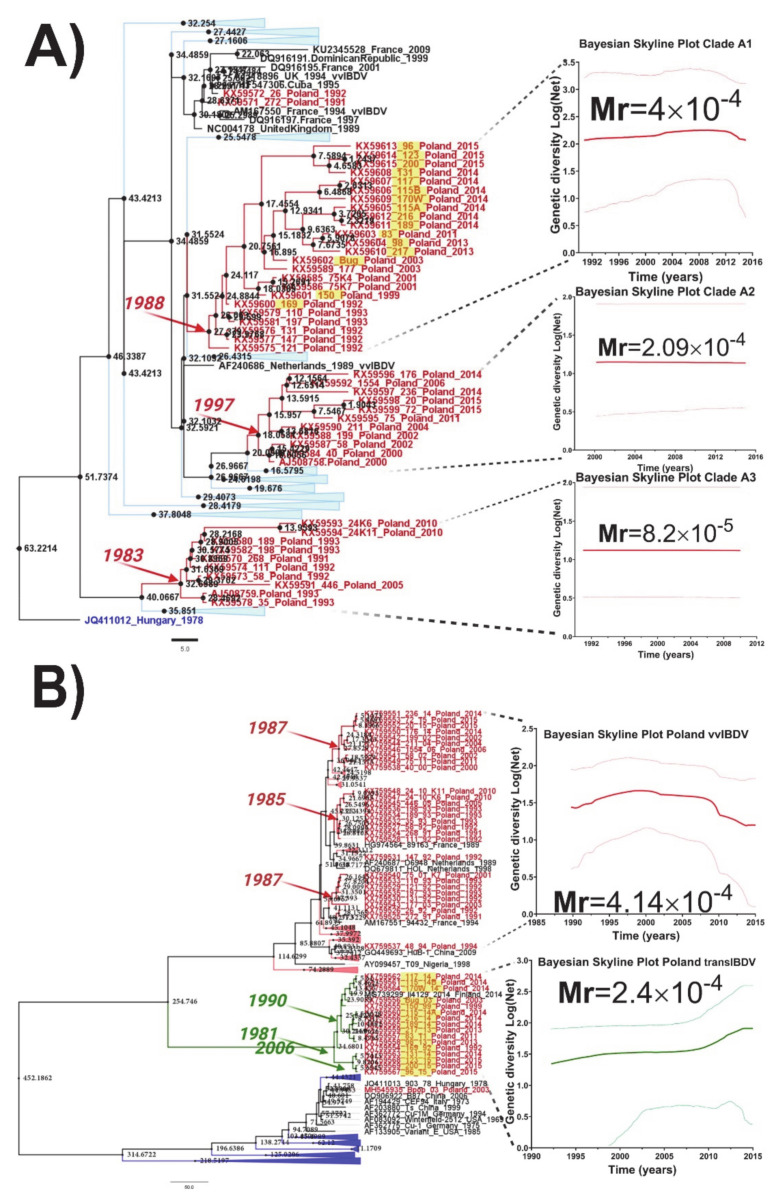
Evolutionary history and population dynamics reconstruction of Polish IBDV strains. (**A**) Maximum clade credibility (MCC) tree based on 175 HVR sequences from the G3a genogroup of IBDV (left) clades A1–A3. The Polish VP2 strains were analyzed for the estimation of tMRCA and phylodynamic reconstruction (right) by a Bayesian skyline plot (BSP) using an exponential, uncorrelated clock model. The *x*-axis is in units of years, and the *y*-axis represents the logarithmic scale of Neτ (where Ne is the effective population size and τ is the generation time). (**B**) MCC tree based on 175 VP1 sequences (left), clades identified within the vvIBDV lineage defined by Polish IBDV strains, and the intermediate B-linage recently reported by Pikuła et al. [32] strains were analyzed for estimation of tMRCA and phylodynamic reconstruction (right) by BSP using an exponential, uncorrelated clock model. The *x*-axis is in units of years, and the *y*-axis represents the logarithmic scale of Neτ (where Ne is the effective population size and τ is the generation time). In all cases, Polish IBDV sequences are denoted in red, and the most probable year for the MRCA within each clade or lineage is denoted. The new intermediate B lineage is denoted in green. For simplification, those clades or lineages from sequences that were not directly related with the Polish sequences were collapsed. The rate of substitution per site per year (Mr) is denoted for each clade or lineage. The support of the clades of interest denoted in the figure were supported by a posterior probability value of pp = 1.0.

**Figure 3 viruses-13-00396-f003:**
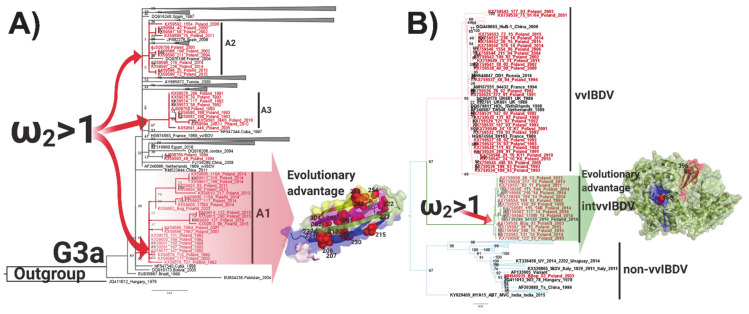
Representation of the topology and identification of positive selected branches and codon sites of Polish IBDV strains. (**A**) Maximum likelihood (ML) phylogenetic tree based on 175 HVR sequences from the G3a genogroup of IBDV clades A1–A3, grouping the Polish VP2 strains, and (**B**) ML phylogenetic tree based on 175 VP1 sequences. In all cases, the clades and lineages that grouped the Polish IBDV strains were analyzed for the branch site model to estimate positive selection using PAML, and the branches that yielded a dN/dS (ω) > 1 are denoted. Polish IBDV sequences are denoted in red. Codon sites found in specific lineages under positive selection (see Appendix A) were mapped on the structure of the HVR of the crystal representation of the VP2 protein (PDB code: 2df7) and VP1 (PDB code: 2pgg) using Pymol. Clades A1–A3 for the Polish strains based on VP2 analysis and all B-lineages are denoted. Arrows represent positively selected lineages that also contain codon sites positively selected, which represent an evolutionary advantage.

**Figure 4 viruses-13-00396-f004:**
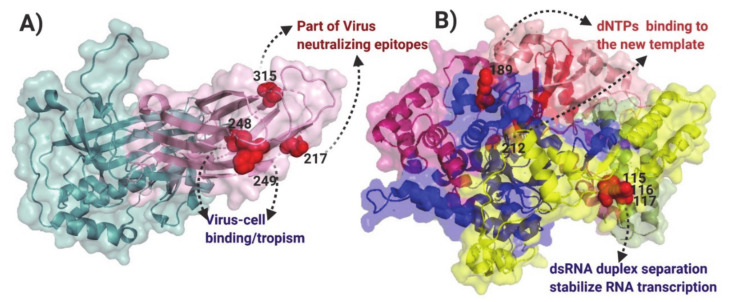
Mapping of positively selected sites and functional divergence sites on the three-dimensional structure of VP2 and VP1 of IBDV. (**A**) Folding of the monomer of the VP2 protein (PBD:2DF7); the shell domain is denoted in aqua and the HVR is denoted in light pink, positively selected sites identified by the site models M2a and M8 vs. M1 and M7, respectively (see Appendix A), are denoted with red spheres, and the functional role for each position under positive selection is also denoted. (**B**) Folding of the monomer of the VP1 protein (PDB:2PGG); the N terminus domain is presented in yellow, the *fingers* domain is presented in blue, the *palm* is denoted in light red, the C terminal domain is presented in hot pink, the functional divergent selected sites are denoted by red spheres, and the functional role for each Type II functionally selected position is also denoted.

**Figure 5 viruses-13-00396-f005:**
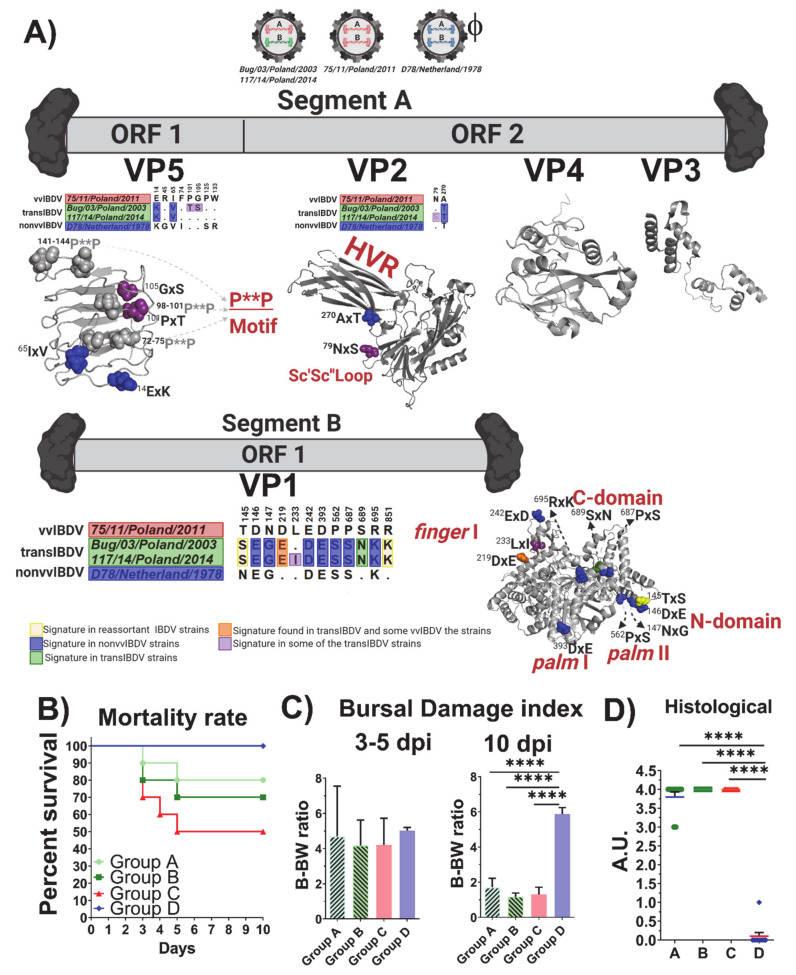
Molecular and biological comparison between the novel IBDV reassortant (vvIBDV-A/transIBDV-B) strains and vvIBDV (vvIBDV-A/vvIBDV-B) strains. (**A**) Schematic representation of the genome characteristic of the novel reassortant strains, the vvIBDV strain used in the experimental infection study, and the attenuated IBDV strains (ϕ only used for comparison purposes in the molecular features) are shown. The representations of both genome segments A and B are presented, all of the molecular signatures found in the ORF encoding regions are denoted in the alignment, and the positions are denoted. All of the amino acid replacements found in the mature protein were denoted in the 3D structure for each protein. All of the amino acid replacements in the 3D structure were denoted using spheres and using the same color code presented in the amino acid alignments. For the VP5 protein, the model generated by [45] and kindly provided by Dr. Ganguly was used, and for the remaining proteins VP2: 2DF7, VP3: 2Z7J, and VP4: 4IZJ PDB, cartons were used. All of the functional domains were amino acid replacements, shown by the novel reassortant strains, are denoted. (**B**) Survival curves of the challenged SPF chickens; different curves indicated different groups, and each group was denoted. (**C**) Estimated bursa damage index determined to 3–5 dpi in the left panel and 10 dpi in the right panel. (**D**) Histological overall scoring. In all cases, significant differences by Brown–Forsythe and Welch ANOVA for multiple comparisons with a Dunnett’s T3 multiple comparisons were denoted (**** *p* < 0.0001). In all cases, the groups are denoted (Group A: animals inoculated with the reassortant strain Bug/03/Poland/2003, Group B: animals inoculated with the reassortant strain 117/14/Poland/2014, Group C: animals inoculated with the very virulent reference strain 75/11/Poland/2011, and group D: control group).

**Figure 6 viruses-13-00396-f006:**
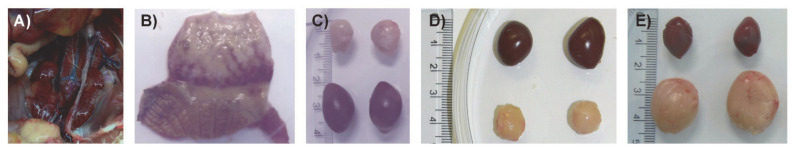
Pathological lesions observed at necropsy in SPF chickens inoculated with the re-assortant strains (Bug/03/Poland/2003 and 117/14/Poland/2014). (**A**) Kidneys from a bird inoculated with Bug/03/Poland/2003 that died at 3 dpi. (**B**) Proventriculus from a bird that died at 5 dpi inoculated with 117/14/Poland/2014. (**C**–**E**) Bursae and spleens from birds euthanized at 10 dpi inoculated with Bug/03/Poland/2003 (**C**), 117/14/Poland/2014 (**D**), and from the normal control (**E**).

**Figure 7 viruses-13-00396-f007:**
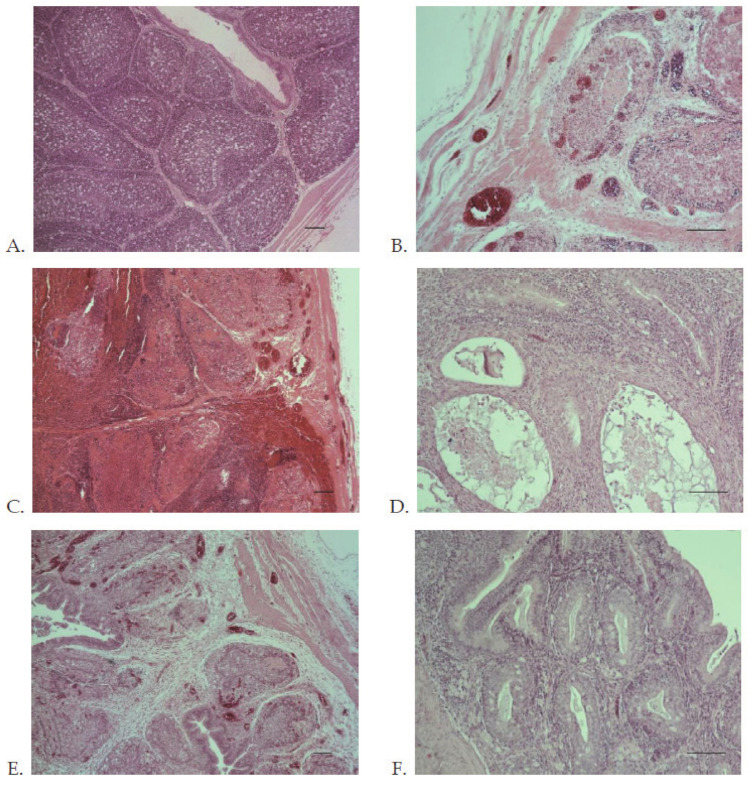
Histopathology of the bursa of Fabricius of SPF chickens. (**A**) The control group at 10 dpi (group D). (**B**) Group C inoculated with reference very virulent strain 75/11/Poland/2011 at 4 dpi. (**C**,**D**) Group A inoculated with Bug/03/Poland/2003 at 3 and 10 dpi, respectively. (**E**,**F**) Group B inoculated with 117/14/Poland/2014 at 3 and 10 dpi, respectively. The magnification is represented by the black band (short = ×50, long = ×100).

**Figure 8 viruses-13-00396-f008:**
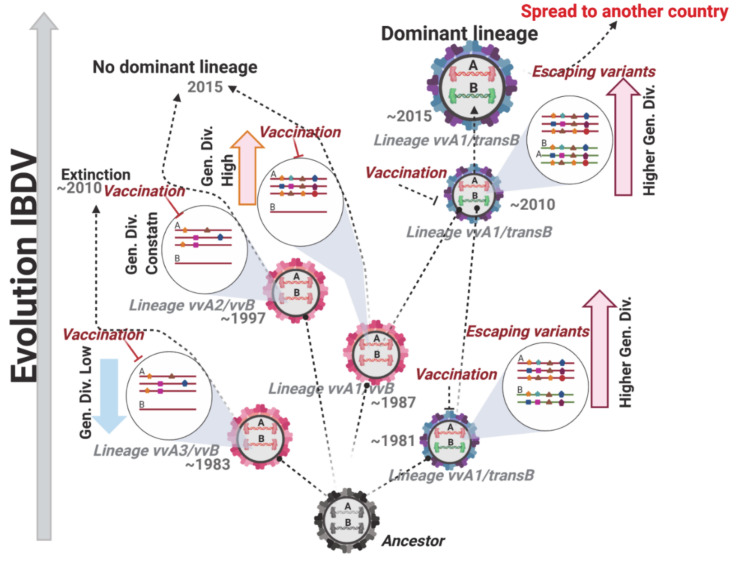
Shift in the dominant genotype strain of IBDV in the field. The evolutionary process of IBDV in the field, year of emergence, and genetic composition for each main lineage are shown. The illustration of quasispecies composition and the bottleneck of different severities induced by the vaccination are also denoted. Effective vaccination is shown in red, and unsuccessful vaccination is denoted in black. The effect of the bottleneck on mutant spectra is shown by denoting relevant adaptive mutations, which are hypothetically located and highlighted with colored symbols.

## Data Availability

The complete genome sequences generated in this study were submitted to the GenBank database (https://www.ncbi.nlm.nih.gov/genbank/) under accession numbers MT629830–MT629835.

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
