# Peer review of "The Novel Genetic Background of Infectious Bursal Disease Virus Strains Emerging from the Action of Positive Selection"

_viruses, 2021, doi:10.3390/v13030396_

Round 1
Reviewer 1 Report
This paper builds on the retrospective results of a study done by Pikula et al. (Transboundary & Emerging Dis. 2020). The results of that study identified Polish IBDV strains and a reassorted IBDV strain containing a vvIBDV genome segment A and a genome segment B from an unknown source (transIBDV-B). In the current study GenBank sequences for 172 VP2 genes and 185 VP1 genes were downloaded and used to examine the evolutionary history of IBDV as it relates to Polish IBDV strains.
The evolutionary analysis of the VP2 and VP1 genes indicated that 3 clades of VP2 and 2 clades of VP1 were present in the Polish vvIBDV strains. The transIBDV lineage was predicted to have emerged by positive selection as were the three VP2 clades. In the last sentence of the Discussion, the authors suggest the vvIBDV/transIBDV reassorted virus could become dominant in Poland. If the transIBDV-B lineage was predicted by tMRCA to have emerged in 1981 with diversification in 1990 and 2006, why is this virus taking so long to become the dominant strain in Polish commercial poultry flocks?
Although the data and data analysis are nicely presented, some of the conclusions are based on assumptions. The authors need to be careful to limit their discussion to their data and relating it to what is actually occurring in nature. For example, the authors indicate the transIBDV has been identified in Finland (2014) as well as Poland. Making assumptions on the origin of transIBDV for these two countries is not realistic without comprehensive survey data. The sequences used in their study came from GenBank and not a targeted survey of countries in Europe. It might be possible the transIBDV is more widespread than just Finland and Poland.
The Discussion is very long. Many paragraphs contain a review of the literature that although relevant to the study, is often not directly related to the data presented. Much of this could be removed without impacting the paper. Specifically, paragraphs starting on lines 496, 508, 526 and 583 could be shortened.
There are many figures in this manuscript. Figures 7 and 8 showing the pathological lesions are not needed. These lesions can be adequately described in the text.
Reviewer 2 Report
Pikula et al. evaluated in depth the evolutionary dynamics shaping the genetic make-up of IBDV strains currently circulating in Poland. They looked at the viruses from multiple points of view and included also results from experimental infections. The manuscript is really interesting, and the study very well performed, including different types of analyses. The manuscript is well written, but it is really, really long and I think the authors should make an effort to cut the text as much as possible to improve clarity and readability. Some supporting analyses can easily be moved to supplementary material without affecting the quality of the study, several redundancies can be eliminated, and several sentences explaining basic concepts about virus evolution can be avoided. In the attached pdf I give some examples of sentences that can be removed, but also other parts of the manuscript can be significantly shortened, and sentences simplified.
Additional comments:
- Introduction. I think it is important to say something about the classification of these viruses as later on in the manuscript you refer to the different lineages (A1, A2, A3) but this has never been introduced. Maybe you can move lines 260-260 to the intro. It would also be important to mention clearly somewhere that this study is focused on genotype 3a.
- The 3D modeling of proteins (or how the figured were obtained) should be added to the M&M.
- Some additional details should be added in the M&M section of the Bayesian phylogeny. Was a modeltest performed to choose the best model for genetic distance calculation and which model was used for the trees? What kind of clock model was chosen (relaxed?) and how was this choice made? Were BF used to compare models? How were sequences aligned?
- Did you remove recombinant sequences from the Beast analyses? If so, this has to be clearly stated, if not this should be done because recombinant sequences affect evolutionary rate and tRMA calculations.
- Figure 4. The figure should show the values for the posterior probability for the clades otherwise the reader can’t have a clear idea of the support for each clade. You can, for example, indicate those instead of the tMRCA at the nodes and include an x axis indicating the years or use colored/shaped labels to indicate PP> a certain value. This should be done at least for the most relevant clades (those used for the BSP and evolutionary rate calculations).
- Lines 305-7. I do not agree with this statement. The difference is minimal and may very well be the results of sampling biases. Please consider removing it.
- Lines 520-525. These are very strong conclusions based on only one mutation whose effect was not even experimentally tested. Also, from your experimental infection experiments the A2 strain caused a higher lethality. Although this is possibly linked to the vvB segment, you did not really prove that, and your results are de facto in contradiction with this statement. Furthermore, most of the changes found in the B segment were found in A1 strain because the reassortant with a very divergent B strain happened with an A1. I suggest rethinking this part and toning down these conclusions because they are very speculative.
- Lines 576-579 and 15-17. I am not sure I can agree with this statement. I see how vaccinations could induce bottlenecks, but the lack of fixation of specific mutations is not necessarily linked to the phenomena described here. The fact that genotype A3 stopped circulating in Poland after 2010 doesn’t mean that it got extinct because of an excessive error rate; it may have simply lost the competition with the other 2 genotypes and the lack of fixation may have been simply due to genetic drift. You also have no proof that any of the mutations are deleterious and the number of sequences available for your analyses is low. I strongly suggest reconsidering this part.
- Figure 9. Types vvA1/vvB and vvA2/vvB could also evolve escape variants. Why did you specifically mention this only for the new reassortant? Also, are there actual mutations that are known to cause immune-escape or this is purely theoretical? Finally, do you have relative lineage frequencies to back up the statement that the reassortant is the dominant lineage? Can you make it clearer that these 2 conclusions are based on speculations (adding ? for example)?
- Lines 731-33. Clinical manifestations are the same, only mortality is a bit lower. I would remove this sentence.
- Lines 580-582. This is also hypothetical. I’d add “potentially” in front of “favoring” because these mutations have not been investigated in vitro.
- Tables 1, 2, 3, 4. All these tables are not essential and they basically repeat results already reported in the manuscript (either in the text or in Figures). Their only function is to show analyses support and they could therefore, easily be moved to the supplementary material.
- The same is true for Figure 2, it can easily be moved to supplementary material as showing this recombination event is only of marginal importance. This would help streamlining the manuscript.
Section 3.4 is really long and can be substantially simplified. All mutations listed in the text are also indicated in Figure 6 and there is really no need to mention them all. Also, all mutation that are in the immature proteins can be left out as you specify at the beginning of the paraph that those are not considered.
Minor:
- Lines 310-312. This should be moved to the discussion section.
- Caption of Figure 4. There is a reference to a “Table XX”. Please, check this.
- Add to the M&M what was used as outgroup in phylogenetic analyses.
- Figure 2. The clades could be labelled more clearly.
- Line 97. This sentence is a bit weird, I suggest rephrasing. Maybe you meant “segments included between breakpoints”?
- Line 135. Specify “likelihood ratio test”.
- Lune 153. There is a typo (“detects”).
Line 196. Weird sentence, please check.
- Figure 4. Please, indicate PP values more clearly, it is really hard to figure out which value belongs to which clade.
- Line 387. Did you mean “immature”?
- Line 449. Did you mean Figure 7?
- Line 452. Typo “Statistically significant”
- Lines 544-549. This sentence is very long and complex. I suggest rephrasing.

Round 2
Reviewer 2 Report
I am satisfied with the revisions and I believe that now the manuscript is much clearer and the readability is improved. However, I still believe that those parts in the discussion section that are speculations/hypotheses are not always clearly labelled as such. This could still be improved.
- Line 118. There is something wrong with this sentence.
- Line 541. Maybe you can add a reference to support your claims since you mentioned in your answer that "The explanation of the roles of the mutations found in both group of strains is very well established in the literature, is well know the mutation found in A3 is linked to escaping of neutralizing antibodies and in A1 is linked to change in virulence"
- The supplementary file does not contain the Supplementary Tables
